# The Effects of Dietary Intervention and Macrophage-Activating Factor Supplementation on Cognitive Function in Elderly Users of Outpatient Rehabilitation

**DOI:** 10.3390/nu16132078

**Published:** 2024-06-28

**Authors:** Yoko Uchiyama-Tanaka, Hajime Yamakage, Toshio Inui

**Affiliations:** 1Yoko Clinic, 3-3-13 Takami, Yahatahigashiku, Kitakyushu 805-0016, Fukuoka, Japan; 2Satista Co., Ltd., 77-1, Minamiochiai, Makishimama-cho, Uji 611-0041, Kyoto, Japan; yamakage@satista.jp; 3Inui Clinic, 3-34, 8-2, Okubo-cho, Moriguchi 570-0012, Osaka, Japan; inuicl@yahoo.co.jp

**Keywords:** dementia, cognitive function, macrophage, AGEs, amyloid beta

## Abstract

Background: Age, genetic, and environmental factors are noted to contribute to dementia risk. Neuroplasticity, protection from degeneration and cell death, and early intervention are desirable for preventing dementia. The linkage between neurons and microglia has been a research focus. In this study, we examined the effects of dietary modification (a reduction in advanced glycation end products [AGEs]) and macrophage-activating factor (MAF; a macrophage regulator) supplementation on cognitive function in elderly participants undergoing rehabilitation. Methods: Participants were older than 60 years of age and had been attending a daycare rehabilitation facility for at least three months without cognitive dysfunction, severe anemia, terminal cancer, or neurodegenerative diseases such as Parkinson’s disease. The exercise protocol at the rehabilitation facility was not changed during the study period. Forty-three participates were randomly divided into three groups: a control group receiving placebo, a group receiving dietary guidance, and a group receiving dietary guidance and MAF supplementation. The amyloid-β40/42 ratio, dietary AGE intake, plasma AGE levels, dietary caloric intake, and mild cognitive impairment (MCI) screen test were evaluated. Results: Four participants withdrew from the study. MCI screening scores significantly improved in the MAF supplementation group, especially after 6 months. Dietary modulation was also more effective than placebo at improving cognitive function after 12 months. Only the control group exhibited significantly increased plasma AGEs while the dietary modulation and MAF supplementation groups showed no change in plasma AGEs after 12 months. Conclusions: MAF supplementation improved cognitive function, especially after 6 months, in elderly people undergoing rehabilitation. Dietary modulation was also effective for improving cognitive function after 12 months compared to that in the control group. It was difficult to supervise meals during dietary guidance at the daycare service. However, simple guidance could show improvements in cognitive function through diet.

## 1. Introduction

Multiple factors have been identified as associated with dementia risk, including age, genetic factors, and environmental factors (lifestyle, exercise, toxin exposure, diet, nutrition, social involvement, hearing, and educational history) [1,2,3]. These factors are believed to trigger protein misfolding, synaptic dysfunction, reactive oxygen species, mitochondrial dysfunction, impaired calcium regulation in cytoplasm, and decreased cell number, resulting in neurodegeneration [1,4,5]. With ageing of the population and an increase in the prevalence of diabetes and atherosclerosis, the prevalence of dementia is on the rise. The Global Burden of Disease (GBD) Dementia Forecasting Collaborators indicate that the prevalence of dementia will reach 150 million by 2050 [6]. If mild cognitive impairment is included in this estimation, the number will be even greater.

Dementia was once thought to be a progressive disease in which cognitive function could not be improved or recovered. However, there are also emerging reports that improvements in the causes of brain inflammation and amyloid-β accumulation may improve symptoms even after a diagnosis of dementia [1,7,8]. Once dementia is diagnosed, organic changes in the brain progress, and recovery is, however, difficult. Therefore, preventing the onset of dementia is highly important. Exercise, a healthy diet, social stimulation, intestinal environment improvement, lifestyle modification, optimal sleep, stress care, hearing aids, nutritional supplementation, and other supplements have been reported to be effective in dementia prevention [1,7].

Notably, brain inflammation is involved in cognitive dysfunction. The linkage between neurons, microglia, and astrocytes has been noted in brain synapse loss, the removal of waste products such as amyloid-β, the control of infection and inflammation, synapse germination, neurogenesis, and pruning [4,5,9,10,11,12,13,14]. Microglia were previously thought to be structural factors in the brain but have been shown to have macrophage functions [4,12]. Microglial dysfunction is a problem in degenerative brain diseases [4,5,10,11]. Enriched protein homeostasis, inflammation, and metal ion homeostasis pathways were observed in astrocytes in animal studies of Alzheimer’s disease. Phagocytosis, inflammation, and proteostasis pathways were enriched in tissue amyloid-rich microglia and perivascular macrophages. The diversity of glial transcription in Alzheimer’s and the response of both astrocytes and microglia to pathological protein clearance and inflammation are associated [15]. In another animal model with neuroinflammatory diseases, Bonneh-Barkay et al. showed that cytokines released from macrophages elevated YKL-40, a marker of inflammation, in astrocytes. This phenomenon is observed among components in the CNS environment. The network of peripheral macrophages, centrally located perivascular macrophages, astrocytes, and microglia may, therefore, be important. Activating ageing macrophages may be important for reducing inflammation and waste accumulation [16].

Group-specific component macrophage activation factor (GcMAF) is a protein that results from the sequential deglycosylation of its precursor, vitamin-binding protein (VDBP) [17,18]. The Gc protein VDBP is produced in the liver and is present in the majority of biological fluids. VDBP has multifunctional properties as a transporter of serum vitamin D3 and its metabolites, functions as an actin scavenger during cellular injury, acts as a chemotaxin for phagocytic cells, and plays a role in macrophage activation as a precursor for GcMAF. This Gc protein has a triple-domain modular structure, where domain III (the C-terminal end) harbors a single glycosylation site. The terminal N-acetylgalactosamine (GalNAc) moiety in domain III is the region involved in the GcMAF-mediated macrophage activation cascade. During inflammation, lysophosphatidylcholine is released from tissue, which induces the expression of beta-galactosidase in B cells and sialidase in T cells. These enzymes hydrolyze the terminal galactose and sialic acid saccharides of the Gc protein to convert it into GcMAF with an N-acetylgalactosamine moiety [17,19]. One of the cell types targeted is resident intestinal macrophages, which exhibit great phagocytic activity without initiating an inflammatory response due to their low, or even absent, expression of innate response receptors, including receptors for lipoplysaccharides (LPS) (CD14), Fcα (CD89), Fcγ (CD64, CD16), and CR3 (CD11b/CD18). Resident intestinal macrophages constitute the largest pool of macrophages in the body. These cells are able to downregulate an excessive systemic inflammatory response by driving the resolution of inflammation and by contributing to a mechanism of oral tolerance to foreign antigens, as well as autoantigens.

MAF was discovered to be a macrophage activator but has also been shown to regulate macrophage function, improve allergies, and optimize immune function, for example, providing adjuvant therapy against cancer and many other diseases [17]. Furthermore, in animal experiments with MAF, recovery from brain damage (embolization model [20] and trauma model [21]) and a reduction in amyloid-β have been observed, and clinically, the effectiveness of MAF has been shown in autism spectrum disorder [22] and multiple sclerosis [23], where brain inflammation is suggested. However, the effects of MAF on cognitive function have not yet been studied.

In examining the effects of supplements, it is important to first consider diet. There are many reports that dietary improvements improve cognitive function, and among these, reports on advanced glycation end products (AGEs) and cognitive function have attracted increasing amounts of attention. AGEs are formed by nonenzymatic reactions between sugars and amino groups of macromolecules, such as proteins, lipids, and nucleic acids [24]. Because glycation occurs constantly during normal ageing and the rate of degradation of glycated macromolecule derivatives is very slow, AGEs accumulate with increasing age. Accumulating evidence shows that AGEs play important roles in the development of age-related disorders, including diabetic vascular complications, osteoporosis, and Alzheimer’s disease [24,25,26,27]. It is estimated that 6–7% of AGEs are taken up by the body from exogenous sources (diet, tobacco, physical activity, etc.) [28,29].

In this study, we examined the effects of nutritional guidance (reducing the amount of AGEs in the diet as much as possible) and MAF supplementation on cognitive function in a group of elderly individuals attending a day rehabilitation facility (that is, elderly individuals who were able to exercise regularly). This study was double-blind with respect to the drug intervention but open with respect to the presence or absence of dietary guidance.

## 2. Methods

### 2.1. Study Design and Participants

This was a single-center, randomized controlled, open-label, parallel-arm trial. Individuals older than 60 years of age who were not diagnosed with dementia, were in stable condition, and had been attending Soushinkai, a daycare rehabilitation facility in Okayama, for at least three months were eligible for inclusion. The procedures for human experimentation were conducted in accordance with the guidelines of the Declaration of Helsinki (2000), and informed consent was obtained from all subjects.

At the time of entry, interviews, evaluations, blood tests, and medical examinations were conducted. Patients with cognitive dysfunction, severe anemia, terminal cancer, or neurodegenerative diseases such as Parkinson’s disease were excluded. Soushinkai rehabilitation facilities provide not only passive rehabilitation but also voluntary rehabilitation, visitation, and assistance so that rehabilitation can be performed at home. The facility also has a daycare center and exercise machines.

Between May 2022 and July 2022, a total of 43 people (mean age 79.0 ± 8.8 years, 17 men and 26 women, care level of support to care level 3) participated in the study. Of these, 15 were allocated to Group C, a normal rehabilitation-only group as a control group; 14 were allocated to Group D, a dietary guidance group, who received dietary guidance and rehabilitation by the same physician; and 14 were allocated to Group M, who received one capsule of MAF triple in the morning upon waking and one capsule before bedtime, and the same dietary guidance as Group D (Figure 1). Groups C and D received placebo in the same manner, with one capsule upon waking and one capsule before bedtime.

MAF triple capsules are dietary supplements produced by Saisei Pharma, Osaka, Japan, which are designed to modulate the mucosal immunity of the intestine. This facility teaches exercise according to the level of care, and the randomization was statistically valid for all three groups in terms of the level of care. Additionally, the patients had to have been visiting the facility for at least three months, be in stable condition, and not have changed their protocol of exercise during the study period unless there was a special reason for doing so.

Before assessing the mild cognitive impairment screening (MCIS) score, AGE levels were measured with an AGE reader. The plasma amyloid-β 40/42 ratio, general blood parameters (peripheral blood, glutamate oxaloacetate transaminase (GOT), glutamate pyruvate transaminase (GPT), gamma GPT, alkaline phosphatase (ALP), hemoglobin (Hb) A1C, blood urea nitrogen (BUN), creatine (Cr), uric acid, total cholesterol, high-density lipoprotein (HDL) cholesterol, triglycerides, ferritin, 25OH vitamin D, C-reactive protein (CRP), free thyroxine (T)4, thyroid-stimulation hormone (TSH), albumin, zinc, copper, sodium, potassium, chloride, folic acid, vitamin B12, and vitamin B1), and dietary AGE intake were measured. The same procedure was used after 6 and 12 months. This study was approved by the Saisei Mirai Ethics Committee (No. 2022-001).

### 2.2. Randomization

After screening, the data of each participant were linked to the participant identification code provided and anonymized. Next, the participants were randomized to Groups M, D, and C at a ratio of 1:1:1. Randomized allocation was performed via the simple randomization method. The allocation table was generated and maintained by a third party at the facility who was not involved in the study.

### 2.3. Dietary Guidance

Dietary guidance was provided to reduce the amount of AGEs in the diet as much as possible. It is generally known that AGEs are produced in large amounts when meat products and fat-rich foods are fried or baked at high temperatures. On the other hand, cooking methods such as slow steaming or boiling of foods with a high-water content are less likely to produce AGEs, which have long been known to be produced in Maillard reactions or browning reactions. The browning of foods is a rough indication of their AGE content. Fructose is approximately 10 times more likely to form AGEs than glucose. It is thought that avoiding drinking carbonated beverages that contain a large amount of fructose corn syrup, avoiding excessive consumption of fructose, and avoiding consuming foods with significant browning can reduce the amount of exogenous AGE intake [30]. Lemon and vinegar can also reduce the formation of AGEs during the cooking process. A well-balanced diet rich in vegetables, seaweed, and mushrooms is also recommended [31].

First, all groups were asked to describe their meals for one week, or if they were unable to do so, they were asked to do so with the help of their family members, who were interviewed in more detail by a nutritionist. Calorie calculations were then performed. AGE calculations were performed with the exAGE formula devised by the AGE Research Association. These calculations were performed before the study and at 6 months. For Group D and Group M, at the start of the study, materials were first distributed to the group, and dietary guidance was given by the same physician in the form of a lecture. The same doctor showed them pictures and instructed them to eat less processed meat (sausage, bacon, etc.), packaged foods, deep-fried foods, sugary sweets, and snacks, and more steamed foods, fresh vegetables, and fruits. Participants were asked to take home the materials used in the information sessions and share the information with their household members. Afterward, during the individual interviews, we provided advice on what to maintain and what to add to the diet of each person based on a dietary chart. Individual interviews were held at the start of the study, at 6 months, and at 12 months. In addition to the doctor’s interviews and check-ins, the dietitian interviewed them monthly to ensure that the dietary instructions were being followed and that they were doing well.

### 2.4. Calculation of AGEs in the Diet

The carboxymethyllysine (CML; a type of AGE) coefficient is determined by dividing foodstuffs into 18 categories, including grains, meats, fats, oils, and vegetables, and multiplying the coefficient of cooking (one for raw foodstuffs and one for boiled or baked foodstuffs) by the coefficient of each category. The following original formula is used to estimate the CML (AGE) content of food products:

CML content (in kilograms) = total protein and fat content (g) per 100 g of food material × processing factor × cooking factor × CML factor. This formula was derived by determining the protein and fat contents of ingredients from the Japanese Food Composition Table. There was a strong positive correlation (*p* < 0.0001) between the measured CML content in foods reported by Uribarri et al. and the estimates obtained in the four factors described above, with a correlation coefficient of 0.52. The average CML intake for the American population is reported to be approximately 15,000 KU [29].

### 2.5. Cognitive Function Assessment

The MCIS was used because it is applicable for those with dementia or mild dementia. The MCIS, derived from the Word List Memory test, differentiates cognitive changes associated with normal ageing from mild cognitive impairment as well as dementia [32]. The MCIS is a brief, electronically scored, verbally administered test that uses correspondence analysis to calculate an individual’s memory capabilities, which is then reported as a Memory Performance Index [33,34,35]. The MCIS has been validated in both academic and community clinical settings and in multiple languages [36,37]. Testing was performed before the start of the study and at 6 and 12 months. Since large social fluctuations in participants have a direct impact on the score, this index is best for use in studies of populations without large social fluctuations.

### 2.6. AGE Measurement

Glycation levels were measured on the dominant volar forearm using an AGE Reader mu (Diagnoptics Technologies B. V., Groningen, The Netherlands) [38,39]. Pentosidine and crossline have structural properties that cause them to emit fluorescent light across a specific range of wavelengths upon excitation by ultraviolet light. This unique characteristic has been used to develop technology that quantifies the accumulated AGEs within human skin. Although several confounding factors exist, such as other fluorophores, skincare cream use, and skin pigmentation, skin autofluorescence (SAF) is a prominent biomarker that may reflect tissue accumulation of AGEs [28,40,41]. Certain types of AGEs autofluoresce when exposed to ultraviolet light. The AGE Reader illuminates skin with ultraviolet light (excitation range = 300–420 nm) and then detects the resulting fluorescent light (emission range = 420–600 nm) while simultaneously detecting light reflected from the skin in the 300 to 420 nm range. The SAF, reported in arbitrary units, is defined as the ratio of the intensity of the emitted fluorescent light to that of the reflected light [29,42,43].

### 2.7. Plasma Ratio of Amyloid-β40 to Amyloid-β42

Recent technological advancements in mass spectrometry have led to improvements in instrument sensitivity and precision, resulting in the development of improved plasma amyloid-beta (Aβ) assays. Many studies have reported encouraging results for plasma Aβ as a biomarker for Alzheimer’s disease. Tests for the risk of developing dementia were conducted at the starting, 6-month, and 12-month time points [44].

### 2.8. Sample Size

The study protocol was designed to detect a large effect size (effect size = 0.8) between groups for the change in endpoint over the 12-month study period. Based on that assumption and for a power of 80% and a 5% significance level, the sample size was calculated to be 13 participants per group.

### 2.9. Statistical Methods

All of the data were analyzed based on the intention-to-treat principle. The primary analysis was performed on the full analysis set (FAS), while the MCIS, which is heavily influenced by social variables, was evaluated in the per protocol set (PPS). The absolute standard difference (ASD) was used to assess the balance of patient backgrounds by random assignment, and the ASD was considered balanced if the 95% confidence interval in this population was less than 0.755 as a threshold value that did not cross zero.

The repeated-measures endpoints were analyzed with linear mixed models that included intervention, dummy variables for time, intervention-by-time interactions, and baseline parameters of endpoints as covariates and participants as a random effect. The covariance structure was a completely general covariance matrix. The results are reported as the least squares means with 95% confidence intervals (CIs) at each time point. A *p* value < 0.05 was considered to indicate statistical significance, and all *p* values were two-sided with multiplicity adjustment by the Bonferroni method. All of the statistical analyses were performed using SPSS version 24.0 (IBM Japan, Ltd., Tokyo, Japan).

## 3. Results

A total of 43 people (mean age 79.0 ± 8.8 years, 17 men and 26 women, care level of support to care level 3) participated. Of these, 15 were allocated to Group C, 14 were allocated to Group D, and 14 were allocated to Group M (Figure 1). In Group D, one participant took donepezil hydrochloride after randomization, and this participant was subsequently excluded from the FAS. In Group C, one participant had a myocardial infarction and discontinued use of the facility, and in Group D, two participants voluntarily withdrew from the study. In Group C, three participants experienced major social changes during the study period. Thus, the resulting PPS included 12 participants in Group C, 13 participants in Group D, and 14 participants in Group M.

The clinical characteristics of the participants are shown in Table 1. The ASD was less than 0.755 (the threshold for which the 95% CI for the ASD did not cross 0 in this study), indicating that the baseline characteristics (age, sex, level of care, presence of diabetes, smoking status, history of education, cognitive function according to the MCIS score, AGEs, the amyloid-β 40/42 ratio, AGE levels, and calories) of the three groups were similar.

The results of the analysis of outcomes for the FAS are shown in Table 2. Dietary caloric intake did not change before or after the intervention in any group, and there were no significant group differences in dietary caloric intake during the intervention (Figure 2). Dietary AGEs were unchanged in Group C but decreased significantly in Group D (−2378 [95% CI: −4724 to −32] KU). Dietary AGEs were unchanged in Group M despite the dietary guidance, and there were no significant group differences in intervention effects (Figure 3). There was not much change in diet in Group M, perhaps because some of the residents were having their meals prepared by their housemates, matching meals with their roommates, or having meals served at other daycares. In addition to diet, there were no significant differences in smoking history and exercise load (nursing care level), which have a significant influence on AGEs in this study (Table 1).

Only Group M had significantly improved MCIS scores after 6 months compared to before the study (from 46.65 [43.51 to 49.79] to 55.79 [52.65 to 58.93], a change of +9.14 [4.70 to 13.58]) in the PPS. The MCIS score of Group M also improved significantly compared to that of both Group C and Group D (*p* = 0.035 and 0.008, respectively). Even after 12 months, Group M maintained the improvement in cognitive function (55.71 [52.45 to 58.97]; a change of +9.06 [4.53 to 13.59]). Compared with before the study, Group D exhibited significant improvements in the MCIS score at 12 months (from 46.73 [43.47 to 49.99] to 54.09 [50.37 to 57.81]; a change of +7.36 [2.42 to 12.3]). Compared with those in Group C, the MCIS scores in Group M and Group D improved significantly at 12 months (*p* = 0.010 and 0.0044, respectively) (Figure 4).

The AGE level measured by an AGE reader did not change significantly in Groups M or D at 6 months or at 12 months in the FAS. Only the AGE level of Group C significantly increased at 12 months (from 2.71 [2.53 to 2.88] to 3.08 [2.90 to 3.25]; a change of +0.37 [0.17 to 0.57]). The AGE level in Group C at 12 months was significantly greater than that in Group M and Group D (*p* < 0.001 and 0.005, respectively) (Figure 5).

The plasma amyloid-β 40/42 ratio showed no significant difference at 6 months, but at 12 months, the plasma amyloid-β 40/42 ratio of all of the groups decreased significantly compared to that before the study (Group C, a change of—1.41 [−2.28 to −0.54]; Group D, a change of—1.39 [−2.34 to −0.44]; Group M, a change of—1.28 [−2.18 to −0.38]). There were no significant differences in serum amyloid beta levels between the intervention groups (Figure 6).

Administration of MAF supplements not only prevented but also improved cognitive decline after 6 months, and the effect was maintained after 12 months. However, Group D, in which participants received dietary guidance, showed improvements in cognitive function after 12 months. The significant differences between Groups M and D disappeared after 12 months. However, compared to the control group, the dietary guidance and MAF-treated groups both showed significant differences in cognitive function. Glycation levels tended to decrease in the MAF-treated group, but no significant pre- or posttreatment differences were observed. The control group showed a worsening (i.e., increase in) of AGE levels, with a significant difference between the dietary guidance group and the MAF-treated group. The amyloid-β 40/42 ratio decreased significantly in all three groups after 12 months compared to that at the start of the study. No adverse events occurred as a result of taking MAF supplements or receiving dietary guidance.

## 4. Discussion

In this study, the administration of MAF supplements improved cognitive function. Microglia play complex roles that are both beneficial and detrimental to disease pathogenesis, including engulfing or degrading toxic entities such as amyloid plaques and promoting neurotoxicity through excessive inflammatory cytokine release [4,9]. Aberrations in the normal homeostatic functions of microglia, such as surveillance, synaptic pruning, and plasticity, may also contribute to excessive synapse loss and cognitive dysfunction in patients with Alzheimer’s disease and other diseases [5,14]. Failure to clear apoptotic cells, cellular debris, and toxic proteins, such as amyloid-β, can contribute to inflammation and neurodegeneration [13,14]. Synapse loss is an early hallmark of Alzheimer’s disease and other neurodegenerative disorders and is considered a strong correlate of cognitive decline. Numerous studies have demonstrated that microglia have many physiological, noninflammatory functions that are crucial for the central nervous system and for regulating neuroplasticity, such as driving neural programmed cell death and synaptic plasticity in adults. Microglia have highly dynamic processes and continually survey their local environment. It is estimated that resident microglia scan the entire volume of the brain as one of their homeostatic functions [14]. Other possible functions of resident microglia include trauma site reduction in post-trauma model mice, amyloid-β plaque and phospho-tau tangle reduction, the repair of infarction in animal models, macrophage activation, and VEGF-increasing effects [20].

Macrophages work in both inflammatory and anti-inflammatory directions, but according to research of Uto et al., [20] the activation of anti-inflammatory macrophages increases when inflammation is high, suggesting that macrophage function is not regulated in one direction but rather changes according to the situation. Siniscalco et al. [22] also reported that GcMAF affects the endocannabinoid system, which controls cell-to-cell communication in, for example, greed, pain, immune regulation, emotional regulation, motor function, development, ageing, neuroprotection, cognitive function, and memory. No side effects were reported after taking MAF supplements in this study. These MAF supplements are made from whey and may cause allergic reactions in people with milk allergies. Few side effects have been reported, but a few percent of people complained of diarrhea.

It is recognized that a high degree of glycation is associated with cognitive function [26,27]. The glycation level is related to exercise, smoking status, and age. AGEs essentially increase with age. Therefore, the fact that only the control group showed a worsening of AGE levels may be attributed to the effects of MAF and dietary guidance (Figure 5). However, only Group D showed a decrease in dietary AGEs, while Group M did not (Figure 3). This finding may be because the meals of many participants in Group M were cooked by day service users or family members, and many participants did not cook for themselves.

Unintentionally, this study was conducted with Group C participants receiving an exercise-only intervention, Group D receiving an exercise and dietary modification intervention, and Group M receiving an exercise and MAF intervention. An improvement in cognitive function in the dietary guidance group was not observed until 12 months (Figure 4), and although there was no difference between Group M and Group C in terms of AGE levels in diet (Figure 2), we believe that reducing AGEs in the diet may be helpful in improving cognitive function in the long term.

The amyloid-β40/42 ratio improved in all three groups after 12 months (Figure 6). All three groups underwent rehabilitation, including an exercise program, so there were no significant differences in the amyloid-β40/42 ratio among the three groups after 12 months, but all three groups showed a decreasing trend in the amyloid-β40/42 ratio compared to the pre-study period. Physical exercise in combination with other interventions, including vascular risk management, diet, and cognitive training, has been found to be associated with improved cognitive performance [2,45,46]. Among the largest improvements were those seen in cognitive functions, overall health, and mood status.

Limitations: We originally planned to continue providing dietary guidance after 12 months; however, cooperation with this guidance was difficult for participants and their family members, and following the guidance involved too much work and time. Since dietary guidance was provided throughout the 12-month period, the AGE levels in the diet after 12 months were unknown. Measuring AGEs levels using self-reported dietary intake and skin autofluorescence has limitations and potential confounding factors that may affect the accuracy of the result. The fact that the participants were in a daycare rehabilitation facility and, therefore, did not follow a uniform diet was also a factor in the ambiguous results. In addition, the advanced age of the participants made it difficult for them to continue the study. Since the exercise load was not changed during the study period and the intensity was varied according to the level of care, there were no significant differences between the groups.

In this study, we did not determine genotypes such as APOE type. The number of participants was originally small, but it decreased further due to withdrawal from the study due to infections and a worsening of underlying diseases. In the future, we hope to conduct a large-scale study with determinations of APOE type at residential facilities where meals can be provided under controlled dietary conditions.

This study showed that cognitive dysfunction is multifactorial but that early intervention can prevent its worsening or even improve cognitive function. The importance of macrophages, the effectiveness yet challenges of dietary guidance, and the importance of social involvement were recognized.

## 5. Conclusions

Our results suggest that cognitive impairment may be prevented by dietary modification and MAF supplementation. In the present study, we were able to show that a simple explanation in an outpatient setting without strict dietary control was effective, and we were able to demonstrate effectiveness by administering a supplement without drugs, injections or other burdensome procedures with strong side effects. Although there are many limitations to this study and stronger evidence needs to be obtained from a larger study in a setting with good dietary management, it is significant that the results could be demonstrated with a simple intervention. MAF supplements took six months to show results, while the dietary intervention required a year to show improvement. Improving cognition through diet requires a considerable amount of time. Although supplements and drugs are often judged to be effective over a six-month or three-month period, lifestyle improvements such as diet may have a slower effect, so it was considered necessary to take enough time for the study period. However, it is costly to continue taking supplements, and strict dietary control is difficult to achieve without institutionalization or other circumstances. Therefore, it was thought that improving diet through simple dietary guidance in a relaxed outpatient setting would be convenient for patients due to its cost and simplicity, even if it took a long time to show effects. In addition to diet and supplements, exercise has a significant impact on cognitive function and AGEs. The amount, type, and degree of achievement of exercise load should be considered more precisely in the future. It was thought that it would be ideal to improve inflammation with supplements in early stages and to improve cognition through diet in the long term, preventing cognitive decline throughout life.

## Figures and Tables

**Figure 1 nutrients-16-02078-f001:**
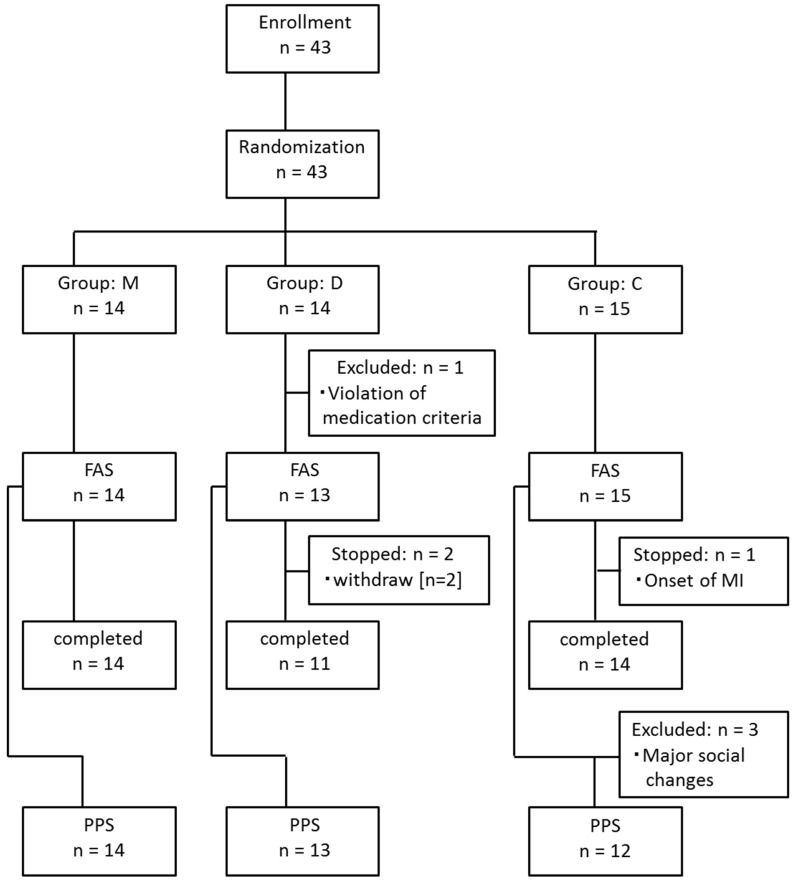
Flowchart of this study. Group M: the MAF triple group; Group D: the diet group; Group C: the control group; FAS: full analysis set; PPS: per protocol set; MI: myocardial infarction.

**Figure 2 nutrients-16-02078-f002:**
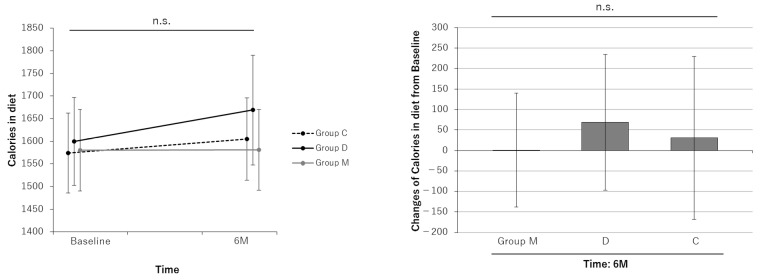
Line and bar graphs of intervention effects on daily caloric intake. Line graphs: Least square (LS) mean and 95% confidence interval (CI) of daily calories at each measurement point for each group are shown. The results of the comparison test between measurement points for each group are shown. Bar graphs: LS means and 95% CIs of daily calories changes from baseline at each measurement point in each group are shown. The results of comparison tests between intervention groups are shown. Analysis method: repeated measures linear mixed model with subjects as a variable [model including initial dependent values as covariates]. Group M: the MAF triple group; Group D: the diet group; Group C: the control group; n.s.: not significant.

**Figure 3 nutrients-16-02078-f003:**
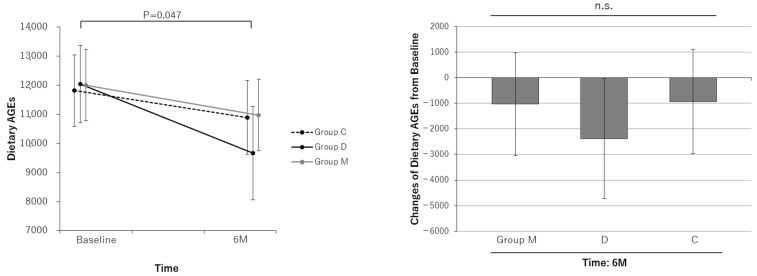
Line and bar graphs showing the effects of the interventions on dietary AGEs. Line graphs: Least square (LS) means and 95% confidence intervals (CIs) of dietary AGE measurements at each measurement point for each group are shown. The results of the comparison test between measurement points for each group are shown. Bar graphs: LS means and 95% CIs of dietary AGE changes from baseline at each measurement point in each group are shown. The results of comparison tests between intervention groups are shown. Analysis method: repeated measures linear mixed model with subjects as a variable (model including initial dependent values as covariates). Group M: the MAF triple group; Group D: the diet group; Group C: the control group; n.s.: not significant.

**Figure 4 nutrients-16-02078-f004:**
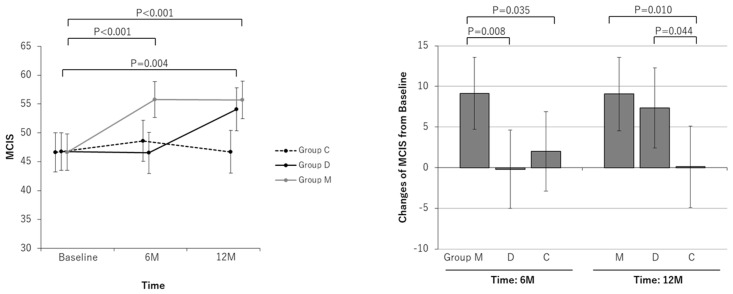
Line and bar graphs of intervention effects on the mild cognitive impairment screen (MCIS). Line graphs: Least square (LS) means and 95% confidence intervals (CIs) of the MCIS measurements at each measurement point for each group are shown. The results of the comparison test between measurement points for each group are shown. Bar graphs: LS means and 95% CIs of MCIS changes from baseline at each measurement point in each group are shown. The results of comparison tests between intervention groups are shown. Analysis method: repeated measures linear mixed model with subjects as a variable (model including initial dependent values as covariates). Group M: the MAF triple group; Group D: the diet group; Group C: the control group.

**Figure 5 nutrients-16-02078-f005:**
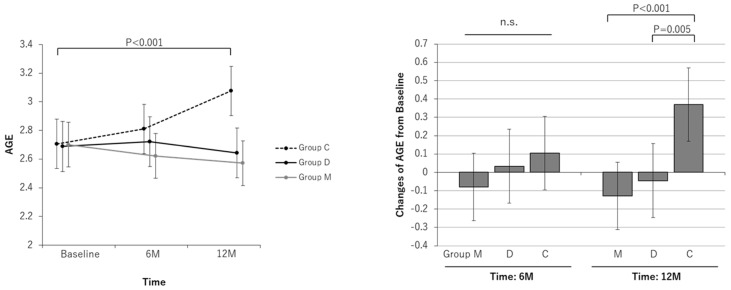
Line and bar graphs of the effects of the intervention on advanced glycation end products (AGEs). Line graphs: Least square (LS) means and 95% confidence intervals (CIs) of AGE measurements at each measurement point for each group are shown. The results of the comparison test between measurement points for each group are shown. Bar graphs: LS means and 95% CIs of AGE changes from baseline at each measurement point in each group are shown. The results of comparison tests between intervention groups are shown. Analysis method: repeated measures linear mixed model with subjects as a variable (model including initial dependent values as covariates). Group M: the MAF triple group; Group D: the diet group; Group C: the control group; n.s.: not significant.

**Figure 6 nutrients-16-02078-f006:**
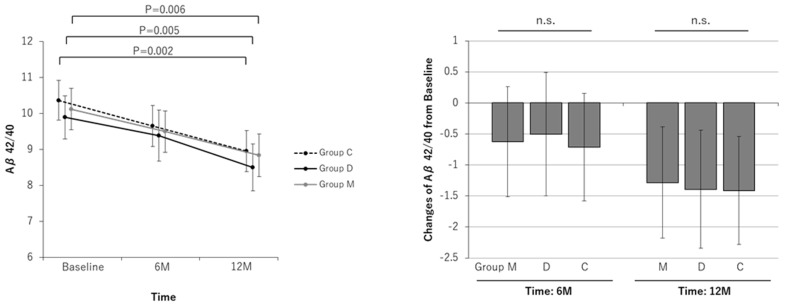
Line and bar graphs of intervention effects on the plasma amyloid beta 40/42 ratio (Aβ40/42). Line graphs: Least square (LS) means and 95% confidence intervals (CIs) of the Aβ40/42 measurements at each measurement point for each group are shown. The results of the comparison test between measurement points for each group are shown. Bar graphs: LS means and 95% CIs of Aβ40/42 changes from baseline at each measurement point in each group are shown. The results of comparison tests between intervention groups are shown. Analysis method: repeated measures linear mixed model with subjects as a variable (model including initial dependent values as covariates). Group M: the MAF triple group; Group D: the diet group; Group C: the control group; n.s.: not significant.

**Table 1 nutrients-16-02078-t001:** Baseline characteristics in the study.

		Group M	Group D	Group C	ASD
n (in FAS)		14	13	15	
Gender [n, %]					0.224
	Male	5, 35.7%	5, 38.5%	7, 46.7%	
	Female	9, 64.3%	8, 61.5%	8, 53.3%	
Age [years]	mean ± SD	80.8 ± 10.3	78.5 ± 10.1	77.8 ± 6.8	0.342
Years of education [years]	mean ± SD	12.1 ± 1.8	12.7 ± 2.1	12.2 ± 2.2	0.307
Nursing care level [n, %]					0.411
	Support Needed	11, 78.6%	10, 76.9%	9, 60.0%	
	Care Needed	3, 21.4%	3, 23.1%	6, 40.0%	
	Support: level 1	3, 21.4%	0, 0.0%	2, 13.3%	
	Support: level 2	8, 57.1%	10, 76.9%	7, 46.7%	
	Care: level 1	0, 0.0%	1, 7.7%	0, 0.0%	
	Care: level 2	1, 7.1%	2, 15.4%	3, 20.0%	
	Care: level 3	2, 14.3%	0, 0.0%	2, 13.3%	
	Care: level 4	0, 0.0%	0, 0.0%	1, 6.7%	
Diabetes mellitus	yes	2, 14.3%	5, 38.5%	6, 42.9%	0.667
Current smoker	yes	0, 0.0%	0, 0.0%	1, 6.7%	0.379
Baseline parameters					
MCIS	mean ± SD	46.4 ± 16.5	47.2 ± 12.2	44.2 ± 13.9	0.230
AGE	mean ± SD	2.7 ± 0.6	2.5 ± 0.5	2.7 ± 0.5	0.269
Aβ40/42	mean ± SD	10.1 ± 2.0	9.7 ± 1.7	10.5 ± 1.7	0.483
dietary AGEs	mean ± SD	11,914 ± 4632	12,287 ± 7108	11,530 ± 3513	0.135
Calories in diet	mean ± SD	1564 ± 272	1626 ± 297	1550 ± 272	0.266

Group M (M): the MAT triple group; Group D (D): the diet group; Group C (C): the control group. SD: standard deviation; ASD: absolute standardized difference; FAS: full analysis set; MCIS: the mild cognitive impairment screen; AGEs: advanced glycation end products; Aβ40/42: plasma amyloid beta 40/42 ratio.

**Table 2 nutrients-16-02078-t002:** Endpoints analysis using mixed models.

	Group M			Group D			Group C			*p* **		
	LSM	95% CI	*p **	LSM	95% CI	*p **	LSM	95% CI	*p **	M vs. C	D vs. C	M vs. D
MCIS												
Baseline	46.65	43.51 to 49.79		46.73	43.47 to 49.99		46.62	43.22 to 50.01				
At 6-month	55.79	52.65 to 58.93		46.52	42.98 to 50.06		48.61	45.06 to 52.15				
Change at 6-month	9.14	4.70 to 13.58	<0.001	−0.21	−5.03 to 4.60	0.933	1.99	−2.91 to 6.90	0.422	0.035	0.526	0.008
At 12-month	55.71	52.45 to 58.97		54.09	50.37 to 57.81		46.71	42.99 to 50.44				
Change at 12-month	9.06	4.53 to 13.59	<0.001	7.36	2.42 to 12.30	0.005	0.10	−4.93 to 5.13	0.969	0.010	0.044	0.624
AGE												
Baseline	2.70	2.54 to 2.86		2.69	2.51 to 2.86		2.71	2.53 to 2.88				
At 6-month	2.62	2.47 to 2.78		2.72	2.55 to 2.90		2.81	2.64 to 2.98				
Change at 6-month	−0.08	−0.26 to 0.10	0.391	0.03	−0.17 to 0.23	0.740	0.10	−0.10 to 0.31	0.300	0.181	0.616	0.410
At 12-month	2.57	2.42 to 2.73		2.64	2.47 to 2.82		3.08	2.90 to 3.25				
Change at 12-month	−0.13	−0.31 to 0.05	0.164	−0.05	−0.25 to 0.16	0.656	0.37	0.17 to 0.57	<0.001	<0.001	0.005	0.538
Aβ40/42												
Baseline	10.12	9.55 to 10.69		9.89	9.29 to 10.49		10.36	9.81 to 10.92				
At 6-month	9.50	8.92 to 10.07		9.39	8.68 to 10.09		9.65	9.08 to 10.22				
Change at 6-month	−0.63	−1.51 to 0.26	0.163	−0.50	−1.50 to 0.49	0.316	−0.71	−1.58 to 0.15	0.106	0.889	0.754	0.856
At 12-month	8.84	8.25 to 9.43		8.50	7.85 to 9.15		8.95	8.38 to 9.53				
Change at 12-month	−1.28	−2.18 to −0.38	0.006	−1.39	−2.34 to −0.44	0.005	−1.41	−2.28 to −0.54	0.002	0.839	0.975	0.871
dietary AGEs												
Baseline	12,003	10,775 to 13,232		12,043	10,717 to 13,370		11,821	10,592 to 13,050				
At 6-month	10,975	9747 to 12,203		9665	8062 to 11,268		10,893	9620 to 12,166				
Change at 12-month	−1029	−3044 to 986	0.307	−2378	−4724 to −32	0.047	−928	−2968 to 1112	0.363	0.944	0.352	0.383
Calories, daily												
Baseline	1580	1490 to 1670		1600	1503 to 1698		1574	1486 to 1661				
At 6-month	1581	1492 to 1672		1669	1548 to 1790		1605	1514 to 1695				
Change at 6-month	1	−138 to 142	0.980	69	−97 to 235	0.410	31	−168 to 106	0.651	0.765	0.726	0.538

Group M (M): the MAT triple group; Group D (D): the diet group; Group C (C): the control group. MCIS: the mild cognitive impairment screen; AGE: advanced glycation end products; Aβ40/42: plasma amyloid beta 40/42 ratio; LSM: least square mean; 95% CI: 95% confidence interval. Analysis method: repeated measures linear mixed model with subjects as a variable (model including initial dependent values as covariates). *p* *: Pairwise comparison with baseline values. *p* **: Pairwise comparison between intervention groups for changes from baseline values.

## Data Availability

The datasets generated and/or analyzed during the current study are not publicly available considering the privacy of the participants but are available from the corresponding author upon reasonable request.

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
