# Peer review of "The Effects of Dietary Intervention and Macrophage-Activating Factor Supplementation on Cognitive Function in Elderly Users of Outpatient Rehabilitation"

_nutrients, 2024, doi:10.3390/nu16132078_

Round 1

Reviewer 1 Report

Comments and Suggestions for Authors

Title and Abstract

Abbreviations should be avoided in the title and abstract of the manuscript. If abbreviations are necessary in the abstract, they should be written out in full the first time they are mentioned; for example, Mild Cognitive Impairment (MCI). The sample size of 43 participants, which is not divisible by three, raises questions about how the participants were distributed among the groups. The abstract should also briefly state the inclusion and exclusion criteria for the participants to enhance their clarity. Additionally, the statement "Dietary guidance in day-care center rehabilitation was difficult due to conflicts with family members and the use of facilities" requires further explanation regarding its implications. Clarification is also needed on the term "enthusiastic users." The abstract should also specify whether exercise was measured and whether dietary improvement led to increased or decreased calorie intake.

Introduction

The information between lines 58 and 70 of the Introduction appears to be out of context. The objectives of this study are unclear. Some abbreviations used in the Introduction are provided without a prior definition:

AGEs - Advanced Glycation End Products

GBD - Global Burden of Disease

MAF - Macrophage Activating Factor

GcMAF - Group-specific Component Macrophage Activating Factor

VDBP - Vitamin D Binding Protein

GalNAc - N-acetylgalactosamine

LPS - Lipopolysaccharides

These terms should be defined for the first time in the text to ensure clarity for readers.

Material and Methods

Group Allocation: Why did the authors not consider including a fourth group that received both dietary guidance and MAF supplementation? This could provide insight into the combined effects of both interventions.

Dietary Compliance: Ensuring that participants adhered to dietary guidance was challenging. Without controlled meal provision, it is difficult to verify the actual dietary intake and accurately measure the amount of AGEs consumed. For this type of study, it might have been better to conduct a clinical trial with meals provided by researchers, which would make it easier to reliably measure the amount of AGEs consumed.

Sample Size: Although sample size was calculated to detect a large effect size, the small number of participants (13 per group) may still limit the statistical power and robustness of the conclusions.

Follow-up Period: A 12-month follow-up period may not be sufficient to observe the long-term effects of the interventions on cognitive function and AGE levels.

Measurement Limitations: Measuring AGEs levels using self-reported dietary intake and skin autofluorescence has limitations and potential confounding factors that may affect the accuracy of the results.

The authors should provide more information to ensure that these limitations do not undermine the results.

Abbreviations Not Defined in the Text

MCIS - Mild Cognitive Impairment Screening

GOT - Glutamate Oxaloacetate Transaminase

GPT - Glutamate Pyruvate Transaminase

ALP - Alkaline Phosphatase

HbA1C - Hemoglobin A1C

BUN - Blood Urea Nitrogen

Cr - Creatinine

HDL - High-Density Lipoprotein

CRP - C-Reactive Protein

T4 - Thyroxine

TSH - Thyroid-Stimulating Hormone

CML - Carboxymethyllysine

SAF - Skin Autofluorescence

Aβ - Amyloid-beta

FAS - Full Analysis Set

PPS - Per-Protocol Set

ASD - Absolute Standard Difference

CIs - Confidence Interval

Discussion

Instead of providing a detailed description of the role of microglia, it would have been more beneficial to compare the results of this study with other studies that have analyzed the effects of MAF on similar outcomes. This would help to better contextualize the findings and strengthen the argument regarding the efficacy of MAF. The results showed that administration of MAF supplements was associated with improved cognitive function. However, it would be helpful to compare these findings with those of other studies that have investigated the effects of MAF in similar models, both in humans and animals.

The statement between lines 359-365 supports my concern regarding the necessity of having a group that receives both dietary guidance and MAF supplementation. Having such a group would provide a clearer understanding of the combined effects of dietary intervention and MAF supplementation on both cognitive function and AGE levels. Potential confounding factors related to meal preparation and adherence to dietary guidelines were also addressed. I suggest that the authors should consider this in future studies to better isolate and understand the effects of these interventions.

Lines 365-372: Firstly, mention of exercise appears here for the first time, which is a significant oversight. Even if this was unintentional, the involvement of exercise as an intervention should have been clearly described in the methods section. To ensure the validity and reliability of the results, it is crucial for the study design to be transparent about all interventions received by each group. Second, the effect of exercise on cognitive function and AGE levels should be discussed in detail. Exercise is known to influence both cognitive health and AGE levels. Ignoring their potential effects and not discussing them can lead to incomplete or misleading conclusions. I recommend that the authors revise the manuscript to include a detailed description of the exercise intervention in the Methods section and discuss its potential impact on outcomes.

While it is commendable that the authors recognize these limitations, some fundamentally undermine the study’s results:

Dietary Guidance Compliance: The difficulty in ensuring compliance with dietary guidance among participants and their family members raises concerns about the reliability of the dietary intervention results. The lack of uniformity in diet due to the day care setting further complicates the interpretation of the data.

Missing Data on AGE Levels: The absence of AGE-level measurements after 12 months due to discontinued dietary guidance significantly limited the ability to draw conclusions about the long-term effects of the intervention.

Genotype Information: The failure to determine genotypes such as the APOE type means that genetic factors, which could significantly influence the outcomes, were not accounted for.

Advanced Age and Study Continuation: The advanced age of participants made it difficult for them to continue the study, suggesting that the study population may not have been ideal for long-term intervention studies.

Given these significant limitations, the results should be interpreted with caution. The study would benefit from a larger scale with controlled dietary conditions and genetic profiling of participants to validate and expand on these findings.

Conclusion

In my opinion, this conclusion repeats the results rather than offering a deeper interpretation. I suggest that the authors consider the following points for a more impactful conclusion.

Interpretation of Findings: Discuss the implications of the cognitive improvements seen in both the MAF supplement and dietary guidance groups. How do these findings contribute to our understanding of cognitive decline interventions?

Mechanisms of action: Exploring potential mechanisms through which MAF supplementation and dietary guidance may contribute to cognitive improvement. For instance, how might MAF influence neuroinflammation or synaptic plasticity?

Long-term benefits and sustainability: Comments on the sustainability of these interventions. What are the long-term benefits and challenges of maintaining cognitive health through MAF supplementation and dietary modifications?

Clinical and practical implications: Highlights practical implications for clinical practice. How might these findings influence dietary recommendations or the use of supplements in the elderly population?

Future Research Directions: Suggest directions for future research, such as large-scale studies, the inclusion of genetic factors, and controlled dietary conditions. emphasizes the need for further studies to confirm and extend these findings.

Final Remarks

In my opinion, this study is valuable, but it must be highlighted that because of its limitations, it can only provide indications for future research. The authors should be very transparent about everything that occurred during the study and describe in detail all the conditions that might have influenced the results. They should aim to explain their results based on similar studies instead of offering general explanations of the mechanisms involved that could apply to any study. Recognizing the difficulty of implementing these studies, it is important to publish the preliminary data. However, the authors should rewrite the manuscript to make it more transparent and accurate regarding the conclusions.

Comments on the Quality of English Language

Only moderate corrections are needed.

Author Response

We sincerely thank you for your careful peer review and detailed comments. Thanks to these reviews, we were able to revise our paper and make further improvements. Thank you very much.

Title and Abstract

First, we have corrected all of the basic points you made about not using abbreviations in the title and abstract. We have highlighted the corrected parts. We hope you will take a look.

Additionally, recruitment for the study has been difficult and we had originally hoped for approximately 60 people, but due to difficulties in obtaining consent and explaining the study to families, the busyness of the staff, and understanding the volume of the blood tests, we were unable to obtain many participants, so we only included 43 people. However, we are able to inform you that the numbers were randomly assigned in the order in which the consent was obtained and that the randomisation was statistically confirmed to be substantial.

The inclusion and exclusion criteria for the participants are listed in the abstract.

Vague phrases such as 'enthusiastic user' were deleted and only the fact that it was difficult to manage the diet in day-care was mentioned.

The amount of exercise was determined according to the level of care, and the protocol of exercise was not changed during the study period.

Introduction

The abbreviations have been reviewed here and added to the necessary sections.

We have also reviewed the context and rewritten lines 58 to 70, which you pointed out, to explain why this explanation about AGEs is necessary.

Materials and Methods

We essentially have a control group that goes to rehabilitation day-care service (group C), a group that goes to rehabilitation day-care service and is guided to consume a low AGE diet (group D), and a third group that goes to rehabilitation day-care service and is guided to consume a low-AGE diet and is also receiving MAF supplements (group M), as you have pointed out. However, we provided the same dietary guidance to users of day-care services who had family members preparing meals instead of themselves, who lived with others and were unable to prepare their meals freely, who ate at other facilities, or who received meals by home delivery. The third group had no change in AGE levels in their diet, and their diet chart showed that they had not changed. The third group consisted of people who consumed only MAF supplements without changing their diet and who went to a rehabilitation facility. This was a major shortcoming of this study. On the other hand, the advantage was that their diet had not changed, so the overall cohort was divided into three groups: the control group, the diet group and the MAF group. The fact that the improvement in the MCI score of the diet group was no longer significantly different from that of the MAF group in the long term, means that the benefits of diet could be reassessed.

As the results in this study were obtained in a daycare setting, it would have been ideal if the diet had originally been thoroughly controlled in a residential setting, as you noted. However, we were unable to find a co-operative facility, and we will definitely try to achieve in the future.

We would also like to increase the sample size in the future and extend the duration of the trial. The limitations of the AGE measurement have been added to the text.

Discussion

We hypothesize that brain inflammation and neurodegenerative disease are strongly related to perivascular macrophages, which are macrophages in various parts of the body, such as the intestine, astrocytes, and glial cells. Therefore, the relationship between glial cells and macrophages, which are involved in phagocytosis and inflammation in the brain, is important, so we think it is important to describe glial cells. However, as you pointed out, we would like to make the text slightly more concise. Thank you for your advice.

The literature shows the ameliorative effect of MAF on amyloid-β accumulation after stroke and a reduction in the size of the brain damage area, which are discussed, albeit in these animal studies. We would appreciate your review.

Please note the aforementioned comments on the grouping for lines 359-365.

As you pointed out for lines 365-372, the intervention of exercise seems to be very important. However, this paper focused on the effects of diet and MAF supplementation. Therefore, I would like to leave it to other papers to discuss the effects of exercise on cognitive dysfunction.

Since this facility guides exercise according to the level of care, and since the randomisation was statistically valid for all three groups in terms of level of care, we do not believe that there is any difference. Additionally, the patients had to have been attending the facility for at least three months, be in a stable condition, and not have changed the protocol of exercise during the study period unless there was a special reason for doing so. This is why we limited the period to one year. We have added that this to the text.

Once again, regarding strict adherence to dietary requirements, we are considering conducting our next study at a residential facility. However, we are happy to show the interesting result that a simple explanation of the diet for those who live at home and participate in day-care can be effective even without strict adherence.

It is true that we have complicated the interpretation, as you said, and we will be working on this issue in the future.

We have described the shortcomings of this study due to the defects in AGE levels in the diet after 12 months, genetic information such as APOE type, and the limited duration of the study due to the elderly age of the patients.

Conclusion

Thank you very much for your advice. I have rewritten all of the conclusions. I have removed the repetition of the results, and based on your advice, I have described the implications of cognitive improvement with diet and MAF supplementation, the long-term benefits and sustainability, the practical implications, and directions for future research. Once again, thank you for your help.

Once again, I would like to thank you for your very detailed review of this issue.

I have rewritten the manuscript, reiterating the limitations of this research in a transparent manner and referring to the opinions of the other reviewers. I have asked a native English speaker to correct my English. We would be grateful if you could review the manuscript again.

Reviewer 2 Report

Comments and Suggestions for Authors

Thanks for the opportunity to review this paper. Please take no offense from my comments; they are only to address some issues within the paper.

There is a very strange way of citing articles that looks like it is an automatic bottom link.

There is a strange division between sections, so, for example, there is no sample size given for sample size, but there is in Results (228)—Between May 2022 and July 2022, a total of 43 people (....

The methods are generally well described; however, some minor details about the execution of the dietary interventions and the consistency of implementation could be more described. For example, the process of the dietary interventions and especially the participants' adherence to them in how this was enforced on the participants could be better described.

It seems that the differentiated study group x lifestyle or social status could have been written in more detail. As the introduction pointed out, smoking has an impact in this respect.

Although the article mentioned limitations, it would have been useful to list the elements that should be controlled for in future studies of this type.

Certainly, a larger group size could have increased the statistical significance of the studies carried out.

The article should also highlight the importance of the mechanisms of action of MAf and potential side effects.

Overall, the article is certainly an important attempt to present the relevance of good health and dietary care in older age, but improvements in individual dimensions would certainly have improved its reception.

Author Response

We sincerely thank you for your careful peer review and detailed comments. Thanks to these peer reviews, we were able to revise the paper and make further improvements. Thank you very much.

Explanations regarding the consistency of the implementation and execution of dietary interventions have been added.

We have included a description of the sample size in the Methods section, which was previously only included in the Results section. Thank you for pointing that out.

We have added the results of the statistics on smoking status and history of education. There was one smoker in the control group, but all of the others were nonsmokers.

In future studies, we will try to improve the sample size, dietary control on the admission, and thorough grouping. We have included a comment about this as well.

We also added a need to investigate the mechanism of action of MAF and its potential side effects.

We would appreciate it if you could take these considerations into account and review the study.

Reviewer 3 Report

Comments and Suggestions for Authors

The manuscript investigates the effect of dietary advice and MAF supplementation on cognitive function. Although the work is well done and sound, some interpretations, formatting issues or the way the results are presented should be improved before publication.

The introduction on preventing cognitive decline is probably not useful as the paper is about people who already have the disease. The same applies to macrophages and B cells as they are not in the brain (the introduction to MAF should only apply to glial cells).

The role of astrocytes in preventing the death of neurons needs to be considered, at least at the introduction.
Sometimes authors use too many references to base a state.
Paper in press (line 82) should be properly cited (authors and year).
Differences in populations should at least be discussed (e.g. the distribution of males and females in group C is different from the other two) and taken into account when interpreting the results.
Figures should be named in the order in which they appear (1. Fig. 1, 2. Fig. 2, etc.), and it should always be stated that the authors are talking about them (even in the discussion). The same applies to tables.
The discussion (line 363) says that only group D has significant results for AGEs, but both M and D have significant results. Please always refer to figure and table. Putting panels (A, B) in figures might also help interpretation.
In the caption of Figure 6 it says plasma and in the text it says serum (line 317) and it is not the same.

The conclusions are actually a summary of the findings. This information should be included in the results section. In the conclusions, the authors should combine several results to draw new conclusions.

Author Response

We sincerely thank you for your careful peer review and detailed comments. Thanks to these peer reviews, we were able to revise the paper and make further improvements. Thank you very much.

As you point out, astrocytes are very important in brain inflammation. We have surveyed the literature on the relationships among perivascular macrophages, astrocytes and glial cells in the brain. We hope that this reiterates the need for MAFs. Thank you very much for your suggestions.

The references have been reviewed and slightly organised. We cut 4 and 16 from the original reference, but added two papers on astrocytes since the reviewer instructed us to mention astrocytes. Therefore, the total number of references is the same. Please understand. Thank you for pointing this out.

Additionally, the paper in line 82 is incorrect: in preparation, not in publication. This has been corrected.

For the population, the validity of the randomisation has been statistically proven. Overall, those who provided consent to participate in the study tended to be more likely to be female. However, we do not believe that this had any impact on the study, as we divided the population into three groups.

Only group D showed significant differences in dietary AGE levels.

On the other hand, the AGE levels in the body on the skin determined using the AGE reader were significant for both groups M and D.

Since it may be useful to understand the values of AGEs in the diet and AGEs in the body, we would like to include a table or figure as you suggested. We believe it will be easier to understand thanks to your suggestion.

All of them are essentially plasma. The use of the word serum is a mistake, and we have corrected it. Thank you very much.

As you noted, the conclusion was a repetition of the results. Once again, we have reviewed and rewritten the conclusion, accounting for the advice from other reviewers. We hope you will review it again.

Round 2

Reviewer 1 Report

Comments and Suggestions for Authors

The authors have made an effort to improve the issues related to the scientific writing style. However, regarding the content, I still have many doubts about the methodology and the interpretation of the results. The text lacks honesty and transparency. The discussion is very speculative. It seems to me that these results are part of a larger project, and the reader would benefit from having the overall results in a single article. Note that in the current version, the authors mention in lines 96 to 98, "In addition, MAF 96 has been reported to be related to life span through telomere extension (paper in preparation)." Moreover, in the elaborated response, "As you pointed out for lines 365-372, the intervention of exercise seems to be very important. However, this paper focused on the effects of diet and MAF supplementation. Therefore, I would like to leave it to other papers to discuss the effects of exercise on cognitive dysfunction." For the reasons stated, I maintain my position of rejection.

Comments on the Quality of English Language

Moderate editing of English language required

Author Response

Once again, thank you for your careful peer review and recognition of the improvements. I have highlighted in pink the areas that I would like to correct and coordinate with one reviewer who have not yet responded to my first revision. Please understand.

The research on telomeres you mentioned is in preparation, and it was a mistake to consider something that has not yet been published as one of the mechanisms. We will delete this document and refer only to the animal studies to date.

Also, regarding the study on exercise, we apologize for the misleading wording. The exercise protocol is based on the level of care, and the condition of participation is that stable patients who have been attending for at least 3 months and who focus on diet and supplements do not change the amount of exercise they do. Once again, I have highlighted the abstract and methodology that describes about physical activity. We also added one additional article on the relationship between AGEs and exercise, and included the link to exercise in the text. We have also included comments on exercise in the results and future issues in the conclusion.

We have asked a professional proofreading service to proofread the English, and we will continue to do so after the revision. We have attached a certificate of proof for your review.

I would appreciate it if you could review it again.

Reviewer 3 Report

Comments and Suggestions for Authors

All requirements have been successfully met. The manuscript is now ready to be published.

Author Response

Thank you for your prompt re-review.
We also appreciate your acceptance of the revice.